# Predictive Coding for Boosting Deep Reinforcement Learning with Sparse Rewards

## Abstract

While recent progress in deep reinforcement learning has enabled robots to learn complex behaviors, tasks with long horizons and sparse rewards remain an ongoing challenge. In this work, we propose an effective reward shaping method through predictive coding to tackle sparse reward problems. By learning predictive representations offline and using these representations for reward shaping, we gain access to reward signals that understand the structure and dynamics of the environment. In particular, our method achieves better learning by providing reward signals that 1) understand environment dynamics 2) emphasize on features most useful for learning 3) resist noise in learned representations through reward accumulation. We demonstrate the usefulness of this approach in different domains ranging from robotic manipulation to navigation, and we show that reward signals produced through predictive coding are as effective for learning as hand-crafted rewards.

## 1 Introduction

Recent progress in deep reinforcement learning (DRL) has enabled robots to learn and execute complex tasks, ranging from game playing (Jaderberg et al., 2018; OpenAI, 2019), robotic manipulations (Andrychowicz et al., 2017; Haarnoja et al., 2018), to navigation (Zhang et al., 2017). However, in many scenarios learning depends heavily on meaningful and frequent feedback from the environment for the agent to learn and correct behaviors. As a result, reinforcement learning (RL) problems with sparse rewards still remain a difficult challenge (Riedmiller et al., 2018; Agarwal et al., 2019).

In a sparse reward setting, the agent typically explores without receiving any reward, until it enters a small subset of the environment space (the "goal"). Due to lack of frequent feedback from the environment, learning in sparse reward problems is typically hard, and heavily relies on the agent entering the "goal" during exploration. A possible way to tackle this is through reward shaping (Devlin & Kudenko, 2012; Zou et al., 2019; Gao & Toni, 2015), where manually designed rewards are added to the environment to guide the agent towards finding the "goal"; however, this approach often requires domain knowledge of the environment, and may bias learning if the shaped rewards are not robust (Ng et al., 1999).

RL problems often benefit from representation learning (Bengio et al., 2013), which studies the transformation of raw observations of an environment (sensors, images, coordinates etc) into a more meaningful form, such that the agent can more easily extract information useful for learning. Intuitively, raw states contain redundant or irrelevant information about the environment, which the agent must take time to learn to distinguish and remove; representation learning directly tackles this problem by either eliminating redundant dimensions (Kingma & Welling, 2013; van den Oord et al., 2017) or emphasizing more useful elements of the state (Nachum et al., 2018a). Much of the prior work on representation learning focuses on generative approaches to model the environment, but some recent work also studies optimizations that learn important features (Ghosh et al., 2018).

In this paper, we tackle the challenge of DRL to solve sparse reward tasks: we apply representation learning to provide the agent meaningful rewards without the need for domain knowledge. In particular, we propose to use predictive coding in an unsupervised fashion to extract features that maximize the mutual information (MI) between consecutive states in a state trajectory. These predictive features are expected to have the potential to simplify the structure of an environment's state space: they are optimized to both summarize the past and predict the future, capturing the most important

elements of the environment dynamics. We show this method is useful for model-free learning from either raw states or images, and can be applied on top of any general deep reinforcement learning algorithms such as PPO (Schulman et al., 2017).

Although MI has traditionally been difficult to compute, recent advances have suggested optimizing on a tractable lower bound on the quantity (Hjelm et al., 2018; Belghazi et al., 2018; Oord et al., 2018). We adopt one such method, Contrastive Predictive Coding (Oord et al., 2018), to extract features that maximize MI between consecutive states in trajectories collected during exploration (one thing worth noting is that this method is not restricted to specific predictive coding schemes such as CPC). Such features are then used for simple reward shaping in representation space to provide the agent better feedback in sparse reward problems. We demonstrate the validity of our method through extensive numerical simulations in a wide range of control environments such as maze navigation, robot locomotion, and robotic arm manipulation (Figure 4). In particular, we show that using these predictive features, we provide reward signals as effective for learning as hand-shaped rewards, which encode domain and task knowledge.

This paper is structured as follows: We begin by providing preliminary information in Section 2 and discussing relevant work in Section 3; then, we explain and illustrate the proposed method in Section 4; lastly, we present experiment results in Section 5, and conclude the paper by discussing the results and pointing out future work in Section 6.

## 2 PRELIMINARIES

**Reinforcement Learning and Reward Shaping**: This paper assumes a finite-horizon Markov Decision Process (MDP) (Puterman, 1994), defined by a tuple $(\mathcal{S}, \mathcal{A}, \mathcal{P}, r, \gamma, T)$. Here, $\mathcal{S} \in \mathbb{R}^d$ denotes the state space, $\mathcal{A} \in \mathbb{R}^m$ denotes the action space, $\mathcal{P} : \mathcal{S} \times \mathcal{A} \times \mathcal{S} \rightarrow \mathbb{R}^+$ denotes the state transition distribution, $r : \mathcal{S} \times \mathcal{A} \rightarrow \mathbb{R}$ denotes the reward function, $\gamma \in [0, 1]$ is the discount factor, and finally $T$ is the horizon. At each step $t$, the action $a_t \in \mathcal{A}$ is sampled from a policy distribution $\pi_\theta(a_t|s_t)$ where $s \in \mathcal{S}$ and $\theta$ is the policy parameter. After transiting into the next state by sampling from $p(s_{t+1}|a_t, s_t)$, where $p \in \mathcal{P}$, the agent receives a scalar reward $r(s_t, a_t)$. The agent continues performing actions until it enters a terminal state or $t$ reaches the horizon, by when the agent has completed one episode. We let $\tau$ denote the sequence of states that the agent enters in one episode.

With such definition, the goal of RL is to learn a policy $\pi_{\theta*}(a_t|s_t)$ that maximizes the expected discounted reward $\mathbb{E}_{\pi,P}[R(\tau_{0:T-1})] = \mathbb{E}_{\pi,P}[\sum_0^{T-1} \gamma^t r(s_t, a_t)]$, where expectation is taken on the possible trajectories $\tau$ and starting states $x_0$. In this paper, we assume model-free learning, meaning the agent does not have access to $\mathcal{P}$.

Reward shaping essentially replaces the original MDP with a new one, whose reward function is now $r'(s_t, a_t)$. In this paper, reward shaping is done to train a policy $\pi_{r'}$ that maximizes the expected discounted reward in the original MDP, i.e. $\mathbb{E}_{\pi_{r'},P}[R(\tau_{0:T-1})]$.

**Mutual Information and Predictive Coding**: Mutual information measures the amount of information obtained about one random variable after observing another random variable (Cover & Thomas, 2012). Formally, given two random variables $X$ and $Y$ with joint distribution $p(x, y)$ and marginal densities $p(x)$ and $p(y)$, their MI is defined as the KL-divergence between joint density and product of marginal densities:

$$MI(X;Y) = D_{\text{KL}}(p(x,y)\|p(x)p(y)) = \mathbb{E}_{p(x,y)}[\log \frac{p(x,y)}{p(x)p(y)}]. \quad (1)$$

Predictive coding in this paper aims to maximize the MI between consecutive states in the same state trajectory. As MI is difficult to compute, we adopt the method of optimizing on a lower bound, *InfoNCE* (Oord et al., 2018), which takes the current context $c_t$ to predict a future state $s_{t+k}$:

$$MI(s_{t+k}; c_t) \geq \mathbb{E}_{\mathcal{S}}[\log \frac{f(z_{t+k}, c_t)}{f(z_{t+k}, c_t) + \sum_{s_j \in \mathcal{S}} f(z_j, c_t)}] \quad (2)$$

Here, $f(x, y)$ is optimized through cross entropy to model a density ratio: $f(x, y) \propto \frac{P(x|y)}{P(x)}$. $z_{t+k}$ is the embedding of state $x_{t+k}$ by the encoder, and $c_t$ is obtained by summarizing the embeddings of previous $n$ states in a segment of a trajectory, $z_{t-n+1:t}$, through a gated recurrent unit (Cho et al.,

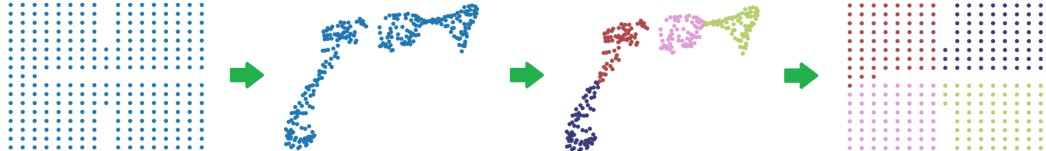

Figure 1: Clustering states in embedding space. We start with the original states (left most), obtain and cluster the embeddings (middle two), and finally label original states by clusters (right most).

2014). Intuitively, the context $c_t$ pays attention to the evolution of states in order to summarize the past and predict the future; thus, it forces the encoder to extract only the essential dynamical elements of the environment, elements that encapsulate state evolution.

## 3 RELEVANT WORKS

Our paper uses the method of Contrastive Predictive Coding (CPC) (Oord et al., 2018), which includes experiments in the domain of RL. In the CPC paper, the *InfoNCE* is applied to the LSTM component (Hochreiter & Schmidhuber, 1997) of an A2C architecture (Mnih et al., 2016; Espeholt et al., 2018). The LSTM maps every state observation to an embedding, which is then directly used for learning. This differs from our approach, where we train on pre-collected trajectories to obtain embeddings, and only use these embeddings to provide rewards to the agent, which still learns on the raw states. Our approach has two main advantages: 1) Preprocessing states allows us to collect exploration-focused trajectories, and obtain embeddings that are suitable for multi-tasking. 2) Using embeddings to provide rewards is more resistant to noises in embeddings than using them as training features, since in the former case we care more about the accumulation of rewards across multiple states, where the noises are diluted.

Applying representation learning to RL has been studied in many prior works (Nachum et al., 2018a; Ghosh et al., 2018; Oord et al., 2018; Caselles-Dupré et al., 2018). In

Figure 2: Illustration of policy trained through clustering. The first-step policy guide the agent towards the correct cluster (white arrow), and the second-step policy guides the agent towards the goal (grey arrow).

a recent paper on actionable representation (Ghosh et al., 2018), representation learning is also applied to providing the agent useful reward signals. In actionable representation paper, states are treated as goals, and embeddings are optimized in a way that the distance between two states reflects the difference between the policies required to reach them. This is fundamentally different from our approach, which aims to extract features that have predictive qualities. Furthermore, computing actionable representation requires trained goal-conditioned policies as a part of the optimization, which is a strict requirement, while this paper aims to produce useful representations without needing access to trained policies.

Lastly, VQ-VAE (van den Oord et al., 2017) is a generative approach that provides a principled way of extracting low-dimensional features. In contrast to VAE (Kingma & Welling, 2013), it outputs a discrete codebook, and the prior distribution is learned rather than static. VQ-VAE could be useful for removing redundant information from raw states, which may speed up learning; however, since the goal of VQ-VAE is reconstruction, it does not put emphasis on features that are particularly useful for learning, nor does it attempt to understand the environment dynamics across long segments of states. Our use of predictive coding is thus be a better fit for reinforcement learning, as we emphasize on features that help understand the evolution of states rather than reconstruct each individual state.

# 4 METHOD

## 4.1 LEARNING PREDICTIVE FEATURES

In this step, the key idea is to train, in an unsupervised fashion and prior to learning, an encoder that extracts predictive features from states. We begin by collecting state trajectories through initial exploration. While there is no requirement for any specific exploration strategies, we used random exploration with manual resets to collect diverse trajectories without need for pre-trained policies.

From these trajectories, segments of consecutive states are sampled and used to train a CPC encoder: for each segment, a fixed number of beginning states $(x_{t-n:t})$ are encoded into latent embeddings $(z_{t-n:t})$ and summarized by a GRU $(c_t = f_{GRU}(z_{t-n:t}))$; the output of the GRU was referred to in the original paper as "context", which is then used to predict the embedding of each remaining state $(z_{t+k})$ in the segment through a score function $s_{t+k} = \exp(z_{t+k} W_k c_t)$. More architectural details can be found in Appendix A.

## 4.2 APPLYING PREDICTIVE FEATURES TO RL

The trained embeddings are then used for reward shaping in 2 ways:

**Clustering**: We sample random states from the environment and cluster their corresponding embeddings. We found that clustering these embeddings provide meaningful information on the global structure of the environments, i.e. states that are naturally close to each other (Figure 1). We then use these clusters to provide additional reward signals to the agent, in particular awarding the agent a positive reward for entering the cluster that contains the goal. This way, the agent is more likely to enter the subset of state space that is close to the goal (Figure 2), and learning is faster and more stable.

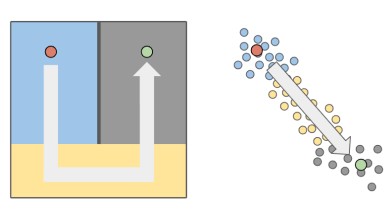

**Optimizing on Negative Distance**: In environments with large state spaces, we directly optimize on the distance between the current state and the goal state in representation space (or embedding space, both of which will be used interchangeably for the rest of the paper). That is, we add an additional negative distance term to the reward at each step $t$, $-\|z_t - z_g\|^2$, where $z_t$ is the embedding of the current state $s_t$, and $z_g$ is the embedding of the goal $s_g$. This simple approach leads to surprisingly good improvements for learning, even for environments with non-linear structures such as mazes (Figure 3).

Figure 3: Illustration of policy trained through optimizing on negative distance in embedding space. Note that moving straight in embedding space (right) may correspond to moving around a wall in the maze (left).

In the next section, we study both the embeddings obtained from initial training as well as both of the applications discussed above.

# 5 EXPERIMENTS

In this section, we will address the following questions:

1. Does predictive coding simplify environment structure?
2. Do these simplified representations provide reward feedback to agent in sparse reward tasks?

We show the result of applying predictive coding to learning in five different environments: Grid-World, HalfCheetah, Pendulum, Reacher, and AntMaze (Figure 4). These environment form a rich set of standard DRL experiments, covering both discrete and continuous action spaces.

GridWorld environment (Chevalier-Boisvert et al., 2018) is used as the primary experiment for discrete settings. Although it has simple dynamics, GridWorld environments have a variety of maze structures that pose an interesting representation learning problem: two points between a wall are close distance wise, but they might require the agent to take a dramatically long route to reach one

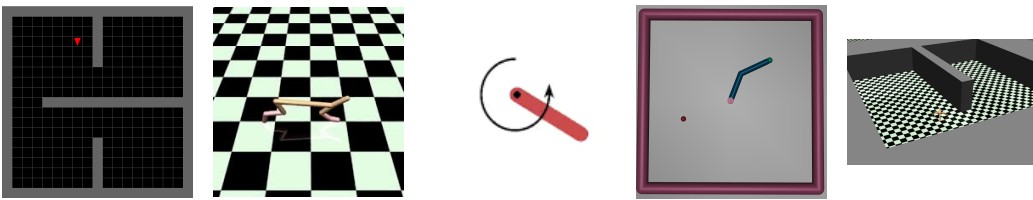

Figure 4: Visualization of environments used in this paper. The environments are GridWorld, HalfCheetah, Pendulum, Reacher, and AntMaze respectively.

point from the other. In our experiments, we demonstrate that predictive coding is able to understand the global structure of any arbitrary maze, and map states in the latent space according to their actual distances in the maze.

We use Mujoco (Todorov et al., 2012) and classic Gym (Brockman et al., 2016) environments for continuous control settings. HalfCheetah, Pendulum, and Reacher are environments in continuous setting with richer dynamics than GridWorld. While they have simpler global structures (e.g. pendulum moves in a circle), we show that predictive coding is able to understand a hierarchy of features in these environments, and that these features can be directly incorporated to speed up learning. AntMaze environments (Nachum et al., 2018b) have continuous control dynamics as well as interesting maze structures similar to GridWorld. As a result, representations learned by predictive coding in AntMaze show both understanding of global structure and formation of hierarchies of features.

To show the generality of our approach, we use an open-sourced implementation of CPC[1] and standard DRL baselines[2] for training. A compilation of all code used in this paper will be made publicly available. Further details on the experimental setup and architectural details can be found in Appendix A.

## 5.1 DISCRETE SETTINGS

### 5.1.1 GRIDWORLD

GridWorld environments are 2D 17-by-17 square mazes with different layouts. We include 3 different layouts: one with a U-shaped barrier (U-Maze), one with 4 rooms divides walls (Four-Room), and one with 4 square blocks (Block-Maze).

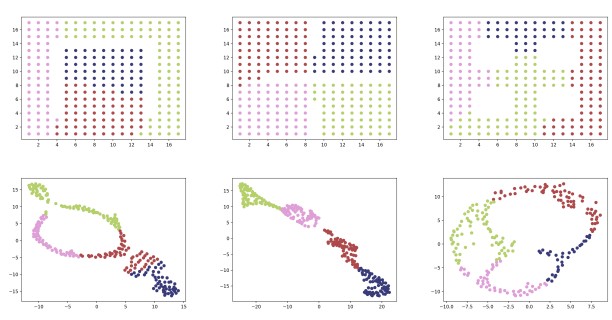

Figure 5: CPC representation of GridWorld environment: U-Maze (left), Four-Rooms (middle), and Block-Maze (right). By clustering embeddings (bottom, all embeddings are visualized by T-SNE), we are able to recover clusters corresponding to natural structures of the mazes (top).

The result of applying CPC representation learning to GridWorld is shown in Figure 5. In all three experiments, CPC learns representations that reflect the true distance between two points. For instance, as seen in the plot, states between the barrier (blue and green states) in U-Maze are mapped to points far from each other in the representation state, although being distance-wise close. Similarly, the states in the blue room and green room of Four-Rooms are mapped to two ends of a long band in the representation space, with states in the red and pink rooms located in the middle. This reflects the need for the agent to go through the red and pinks rooms to reach the blue room or the green

---

[1] https://github.com/davidtellez/contrastive-predictive-coding
[2] https://github.com/hill-a/stable-baselines

room. Lastly, representation learned in Block-Maze restores the true structure of the maze from imaged-based observations.

We assess the quality of the embeddings by analysing how much they reflect the true distances between states. For each maze environment, we sample random pairs of points from the maze, and plot the true distance (obtained by running $A^*$ in the original maze ) between the pair against their distance in the representation space (L-2 norm). Additionally, we run linear regression to obtain a line of best fit for each plot. The result for Barrier Maze is shown in Figure 6, and for all three mazes we observe a strong correlation between the true distances and the distances in latent space.

We show the result of applying clustering to sparse reward problems in GridWorld: the agent is randomly spawn and navigates in the maze, and it only receives a positive reward when reaching the goal. Without additional reward signals, the agent might not be able to reach the goal if it is spawn in a far-away location. To make use of clusters obtained from CPC representations, we train a two step policy: the agent first goes to the cluster that contains the goal, and then to the goal. We reward the agent for reaching the cluster in the first step, and use environment reward for the second step. This way, the agent receives more signal in all locations in the maze. An illustration of a policy trained this way is shown in Figure 2.

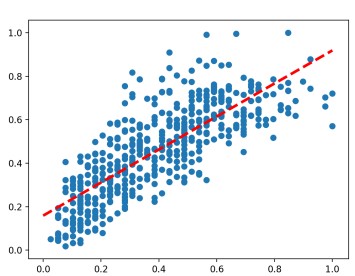

Figure 6: Plot of true distance between states in maze vs distance in representation space. The x-axis is the true distance obtained by $A^*$, while the y-axis is the distance in the representation space. A line of best fit is provided for illustration.

We find that this approach leads to better learning in all three mazes (Figure 7). In all three experiments, the reward (adjusted to remove cluster reward) converges faster and to higher values with clustering. This is likely because the additional reward for entering the cluster guides the agent towards states that are naturally close to the goal, allowing the agent to reach the goal more frequently during exploration. Table 1 shows that the policy learned through clustering significantly outperforms the policy learned in standard setting.

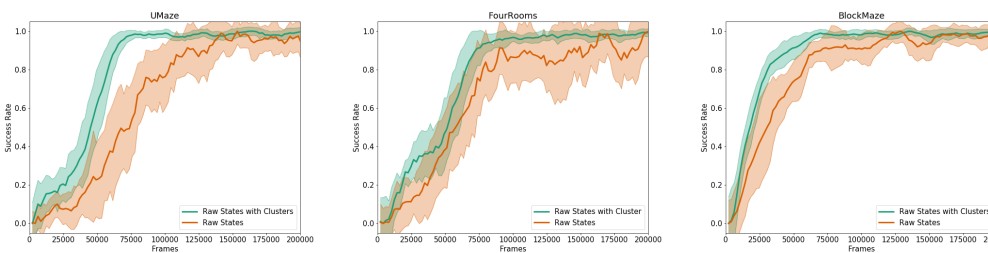

Figure 7: Learning curves of GridWorld environments, averaged over 3 runs. The reward curve with clusters is shifted down by 1 to remove the cluster reward.

Table 1: Success Rate in GridWorld Environments. A successful run indicates that the agent reaches the goal from a random stating potion in the maze within 100 steps.

| Setup | Without Clustering | With Clustering |
|---|---|---|
| U-Maze | 61% | 98% |
| Four-Rooms | 71% | 98% |
| Block-Maze | 98% | 100% |

## 5.2 CONTINUOUS SETTINGS

We study four different control environments: Pendulum, AntMaze, Reacher, HalfCheetah (latter two moved to Appendix A).

Environments used in this section have much richer dynamics than GridWorld. We show that the learned representations simplify these environments both by understanding the global structure of the environment (AntMaze) and encoding meaningful hierarchies of features.

Unlike GridWorld, simple clustering strategies are less effective because of the large state space. Instead, we directly optimize on the agent's distance to goal in representation space. We show that this simple approach can lead to improvement in learning as much as using hand-shaped rewards. For each environment, we include 4 setups, each setup using 3 random seeds:

1. Sparse reward (blue): Providing an agent a small positive reward when it reaches the goal

2. Hand-shaped reward (pink): Providing an agent a hand-shaped reward at each step (hand-shaped reward)

3. Raw distance (green): Providing an agent a negative penalty on the distance between current state and goal state (plus sparse reward to goal for AntMaze only)

4. Embedding distance (orange): Providing an agent a negative penalty on the distance between current state and goal state in the representation space (plus control penalty on action norm for HalfCheetah only).

The details of the hand-shaped reward schemes can be found in Appendix A

### 5.2.1 PENDULUM

The Pendulum is a classic control problem where a rigid arm freely swings about a fixed center. To imitate the swing-up task, we set the goal states to have angles $\theta \in [-0.1, 0.1]$, where an angle of $0$ means the arms is pointing straight up. Additionally, we consider "the goal is reached" only after the agent manages to stay among goal states for 5 consecutive steps. For hand-shaped reward, we penalize the magnitude of the angle encourage the arm to maintain an upward position.

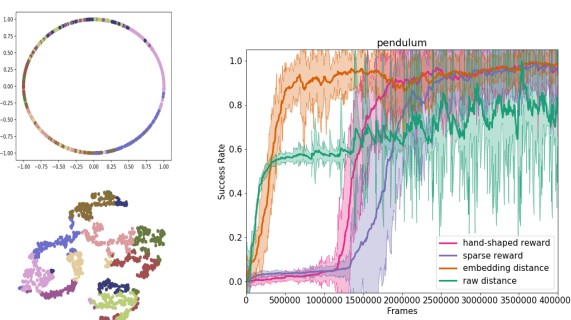

Figure 8: Illustrations of embeddings of Pendulum and learning curves for different reward schemes. Embeddings (bottom-left) cluster primarily by angle of the arm (top-left). Orange curve converges significantly faster than others.

As shown in Figure 8, clustering the embeddings produces clusters primarily by angle. However, there is still lot of overlapping between clusters when we only consider the position of the arm, suggesting that the arm's velocity also plays a less important role. This hierarchy between position and velocity was very beneficial for learning, as the agent would learn to swing up the arm first before decelerating the arm to maintain top positions. Indeed, optimizing on distance in embedding space (orange) led to much faster learning than all other setups, including the hand-shaped rewards (pink), where as optimizing on distance in the original space (green) leads to sub-optimal behaviors such as reducing velocity too early.

### 5.2.2 ANTMAZE

Finally, in AntMaze, a four-legged robot navigate in a maze-like environment until it reaches the goal area. Naturally, AntMaze has both the rich dynamics of a robot as well as the structure of a maze environment, and is helpful for illustrating the power of predictive coding to both reflect the global structure of an environment while picking the most important features. In our experiment setup, we use a thin wall to block passage in the maze, so that a state on the other side of the wall

may appear close to the agent, but is in reality very far. We set the goal state to be the lower left corner of the maze; for hand-shaped rewards, we assign each state in the maze a correct direction to move in and award the agent for moving in that particular direction.

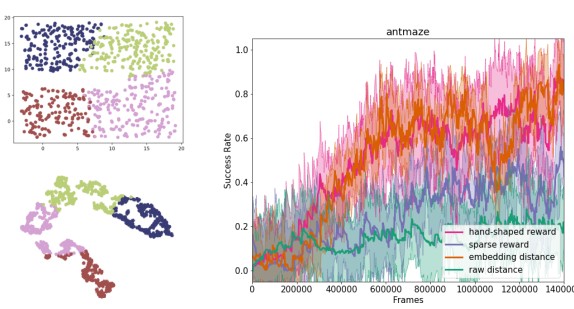

Figure 9 shows the visualization of embeddings and learning curves. One immediate observation is that clusters properly divide the maze into 4 sections, and states between the wall are now more separated; this is similar to the embeddings in Grid-World, where the embeddings understand the true distance between different positions in the maze. At the same time, even though full states are provided (positions as well as full joint dynamics), the clusters reflect that embeddings learn to use the position of the agent as the most important feature. As a result, learning using these embeddings (orange) achieves performance on par with

Figure 9: Illustrations of embeddings of AntMaze and learning curves for different reward schemes. Embeddings (bottom-left) cluster primarily by position of the agent (top-left). Orange curve achieves similar performance to pink curve with smaller variance.
hand-shaped reward (pink).

## 6 DISCUSSION

### 6.1 EMBEDDINGS AS FEATURES VS REWARD SHAPING

Mutual information maximization is notoriously difficult to optimize, and may easily produce noisy embeddings without sufficient training data. Our approach mitigates this problem in two aspects. Firstly, we preprocess the embeddings instead of training them online, so that the agent avoids learning on noisy embeddings that are not fully trained. Secondly, instead of using the embeddings as features to train on, we use them to provide reward signals to the agent, who still learns using the raw features. This approach is more resilient to noises in embeddings, especially for policy gradient methods, since we care more about total rewards across trajectories than the rewards of individual states.

We illustrate the above points by comparing training with cpc features and our approach in the Reacher environment. Both experiments use the same architectural and algorithmic settings, and the raw states used for training the embeddings contain information about the goal. As shown in Figure 10, the use of cpc embeddings as features lead to insignificant improvements to learning, where as using these embeddings to only provide reward signals led to the best performance.

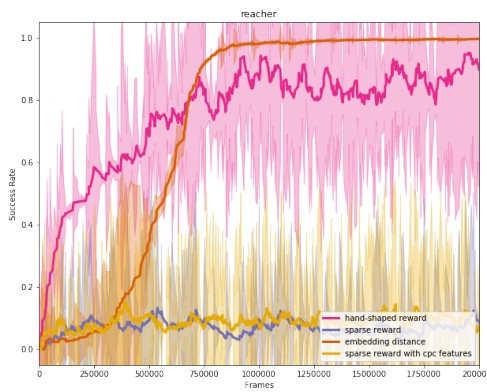

Figure 10: Illustration of learning curves for different setups. Learning on the original sparse reward problem with embeddings as features (yellow) did not lead to significant learning improvement, while using embeddings to provide rewards (orange) achieved the best learning.

### 6.2 TEXTURE AGNOSTIC PREDICTIVE CODING

In this section, we discuss an important advantage of predictive coding: since embeddings are optimized to maximize their predictive abilities, less meaningful information such as the texture of the

background from the raw observations are ignored. This property of predictive coding differentiates itself from other unsupervised learning methods such as autoencoder or VAE, which inevitably pay attention to the background in order to reconstruct the original states.

This property of predictive coding makes it possible for an agent to learn in a constantly changing environment, such as a game (Bellemare et al., 2012). We showcase this property by training the encoder with states from the Pendulum environment with multiple backgrounds (bricks, sand, cloth), and assess the encoder's generalizability to new textures (such as wood). Figure 11 contains examples of textures used for training and validation, as well as the clustering results of their corresponding embeddings. In particular, two textures resulted in very similar embeddings, even though the encoder had never seen the wooden texture during training. We conclude that predictive coding has learned to ignore the background, which contains less important information about the state dynamics.

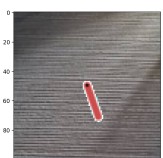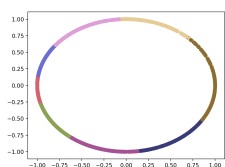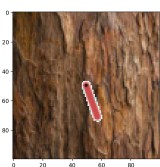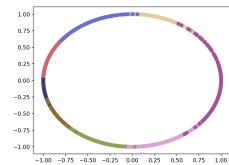

Figure 11: Illustrations of two background textures and clustering of their corresponding embeddings (clustered by embeddings, illustrated by mapping to position of arm). The cloth texture (left) was used during training, while the wood texture (right) was only used during validation. Both textures led to embeddings properly clustered by the angle of the arm.

## 6.3 EXPLORATION

Our proposed method relies on the quality of trajectories collected at the beginning, which in turn depends on the initial exploration. Although in most cases, exploration with random policies or simple goal-conditioned policies is enough to produce trajectories that expose the environment dynamics, there are environments with extremely long horizons or large state spaces that effective exploration without learning the task is difficult. An example is Montezuma's Revenge, which is currently unsolvable without algorithms designed to tackle hard exploration (Ecoffet et al., 2018) or expert demonstration data (Salimans & Chen, 2018). For future work, a direction is to train the embeddings online, i.e. during training the agent. This way, trajectories collected may be more relevant to the particular training task, and we could obtain high-quality embeddings (high-quality in the sense that they are useful for the particular training task) without thorough exploration of the environment. As discussed in the first paragraph, learning on intermediate embeddings may be undesirable, so the agent should initially rely purely on environment rewards, and only start receiving rewards shaped by embeddings after the embeddings reach a certain quality mark (for CPC, this could be checked by the *InfoNCE* loss, which indicates a lower bound on mutual information).

## 6.4 NEGATIVE DISTANCE AS A POTENTIAL FUNCTION

One of the major issues with reward shaping is that it could potentially bias learning, leading the agent to learn a suboptimal policy. In a previous work on policy invariance under reward transformation (Ng et al., 1999), the notion of potential-based reward shaping function establishes the conditions for guaranteeing unbiased learning. With the use of predictive coding, our negative distance reward scheme is a potential function if the latent space induced by the encoder preserves the metric properties of the original state space. A rigorous formulation of this setting necessitates a mathematical analysis, which is out of the scope of this preliminary study.

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

# A APPENDIX

## A.1 ENVIRONMENT DESCRIPTIONS

**GridWorld**: The agent is a point that can move horizontally or vertically in a 2-D maze structure. Each state observation is a compact encoding of the maze, with each layer containing information about the placement of the walls, the goal position, and the agent position respectively. The goal state is one in which the goal position and the agent position are the same.

**HalfCheetah**: The agent is a 2-d two-legged robot freely moving on a horizontal plane. Each state observation encodes the position of the agent, the angles and the velocities of each joint. The goal states are any states where the agent position $x \geq 10$.

**Pendulum**: The agent is a rigid arm moving about a fixed center by applying a force to its tip. Each state observation encodes the angle and the velocity of the arm. The goal states are any states where the agent achieves an angle $\theta \in [-0.1, 0.1]$.

**Reacher**: The agent is a robotic arm with two rigid sections connected by a joint. The agent moves about a fixed center on the plane by applying a force to each rigid section. Each state observation encodes the angles of two sections, the position of the tip of the arm, and the direction to goal. The goal state is one where the tip of the arm touches a certain point on the plane.

**AntMaze**: The agent is a four-legged robot freely moving in a maze structure. Each state observation encodes the position of the agent, the angles and the velocities of each joint. Instead of learning from scratch, we pre-train a simple direction-conditioned walking policy and learn to navigate in this environment. The goal states are a square area with side 2 and center $(0, 0)$.

Table 2: Environment dimensions and horizons

| Environment | State/Goal Dimensions | Action Dimensions | Maximum Steps |
|---|---|---|---|
| GridWorld | (17, 17, 3) | (4,) | 100 |
| HalfCheetah | (18,) | (6) | 1000 |
| Pendulum | (3,) | (1,) | 200 |
| Reacher | (11,) | (2,) | 50 |
| AntMaze | (113,) | (2,) | 600 |

## A.2 NETWORK PARAMETERS AND HYPERPARAMETERS FOR LEARNING

For all experiments in this paper we use a standard PPO baseline (Hill et al., 2018) to train. We use two fully-connected layers with output size 64 for the actor critic. We provide hyperparameters below, and refer to Hill et al. (2018) for all other implementation details.

Table 3: Hyper parameters for PPO

| Parameter | Value |
|---|---|
| gamma | 0.99 |
| entropy coefficient | 0.01 |
| leaning rate | $2.5 \times 10^{-4}$ |
| clip range | $[-0.2, 0.2]$ |
| max gradient norm | 0.5 |
| batch size | 128 |

## A.3 TRAINING CPC

We follow the approach proposed in Oord et al. (2018) to obtain predictive features from states. The details are provided in the two tables below. All hyperparameters for training CPC are tuned through performing grid search.

Table 4: Network details for training CPC

| Environment | Encoder Type | Autoregressive Model Type |
|---|---|---|
| GridWorld | 2 FC layers with output size 64 | GRU with output size 256 |
| All others | 2 Cov layers with (3, 3) kernel, and stride 2 | GRU with output size 256 |

Table 5: CPC training details

| Environment | No. of Trajectories | Length of Trajectory | Context Length | Predict Length | Epoch |
|---|---|---|---|---|---|
| GridWorld | 200 | 100 | 10 | 10 | 2 |
| HalfCheetah | 500 | 1000 | 50 | 50 | 5 |
| Pendulum | 200 | 300 | 10 | 10 | 10 |
| Reacher | 200 | 500 | 20 | 20 | 10 |
| AntMaze | 500 | 1000 | 30 | 30 | 5 |

## A.4 HAND-SHAPED REWARD SCHEMES

For continuous environments, we show that using predictive features provides reward signals as informative as hand-shaped rewards, which encodes domain and task knowledge about the environment. In particular, each hand-shaped reward scheme contains information about where the goal is and how to get there. We provide details below for each environment.

- **HalfCheetah**: $(x_{t+1} - x_t) - \alpha \|a_t\|$, where $x_t$ is the x position of the agent at time $t$, and $a_t$ is the action input to the agent at time $t$.

- **Pendulum**: $-(|\theta_t|^2 + \alpha |\omega_t| + \beta \|a_t\|)$, where $\theta_t$, $\omega_t$, and $a_t$ are the angle, angular velocity, and action input at time $t$ respectively.

- **Reacher**: $-(\|p_t - p_g\| + \alpha \|a_t\|)$, where $p_t$ is the position of the tip of the arm at time $t$, $p_g$ is the position of the goal, and $a_t$ is the action input at time $t$.

- **AntMaze**: $-((x_{t+1} - x_t)\cos(\theta_t) + (y_{t+1} - y_t)\sin(\theta_t))$, where $x_t, y_t$ are the x, y positions of the agent at time $t$, and $\theta_t$ is the hard-coded direction to travel in.

## A.5 EXPERIMENT RESULT FOR HALFCHEETAH

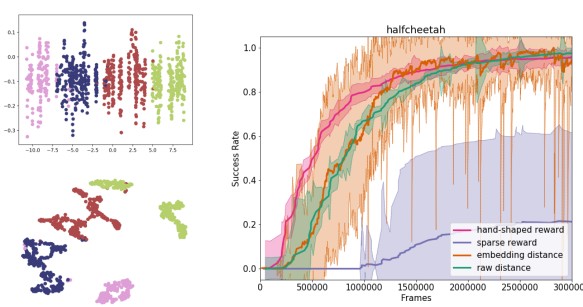

Figure 12 shows the visualization of embeddings of random states as well as the comparison between different reward setups. The predictive features focus on the most significant element of the environment: the x position of the agent, allowing us to recover horizontally spaced clusters.

As the plot shows, optimizing on the negative distance in embedding space (orange), optimizing on the negative distance in raw space (green), and optimizing hand-shaped rewards (pink) all lead to similar performances. While this is less convincing than other environments, we observe that optimizing negative distance in representation space is not

Figure 12: Illustrations of embeddings of HalfCheetah and learning curves for different reward schemes. Embeddings (bottom-left, visualized by T-SNE) cluster by x position of the agent (top-left, with x-aixs being x position and y-axis being y-position of the agent).

*worse*; rather, it is likely that optimizing on negative distance in raw space is already *good enough*, since the x-position of the agent has the largest variance among all other features.

## A.6 EXPERIMENT RESULT FOR REACHER

In the Reacher environment, a robotic arm has two sections with a joint in the middle. The arm's one end is fixed at the center of the plane, and its goal is typically to reach a certain point on the plane with the other end of the arm. This can be naturally formulated as a sparse reward problem (Lanka & Wu, 2018), where the agent receives no reward until it reaches the goal state. For hand-shaped reward, we penalize the distance between the tip of the arm and the goal point plus the L-2 norm of agent's action for stability.

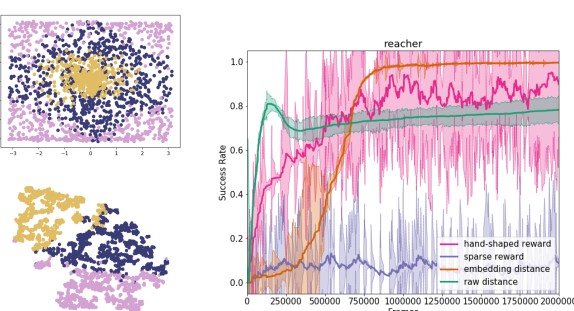

Figure 13: Illustrations of embeddings of Reacher and learning curves for different reward schemes. Embeddings (bottom-left) cluster primarily by angles of sections of the arm (top-left). Orange curve achieves almost-perfect performance at 750k steps.

Similar to Pendulum, embeddings of Reacher achieves clusters primarily by the position of the arm, as shown in Figure 13 (note that each axis is the angle of one section of the arm). Consequently, optimizing on the distance in embedding space (orange) allows the agent to quickly learn to move directly towards the goal. This turned out to be more stable than using hand-shaped reward (pink), which sometimes led the agent to occasionally overshoot and miss the goal.

