# OpenReview forum: "Predictive Coding for Boosting Deep Reinforcement Learning with Sparse Rewards"
_ICLR.cc/2020/Conference — Reject_

### Official Review · AnonReviewer3 · 2019-10-22
**Official Blind Review #3**

**Rating:** 3

**Review:**

The paper proposes a reward shaping method which aim to tackle sparse reward tasks. The paper first trains a representation using contrastive predictive coding and then uses the learned representation to provide feedback to the control agent. The main difference from the previous work (i.e. CPC) is that the paper uses the learned representation for reward shaping, not for learning on top of these representation. This is an interesting research topic.

Overall, I am leaning to reject this paper because (1) the main contribution of the paper is not clear (2) the experiments are missing some details and does not seem to support the claim that the proposed methods can tackle the sparse reward problem.

First of all, it would’ve been better to have a conclusion section, so the readers can see the contributions of the paper. After reading the paper, I still do not understand what are the contributions of the paper and what're from the previous works. The paper does not provide well justification why CPC feature can provide useful information for reward shaping. The paper does not provide a new method to learn predictive coding. It does not provide a novel reward shaping method (the “Optimizing on Negative Distance” method is very similar to [1]). So, I am not sure what’re the contributions of this paper.

Moreover, I am not convinced that the proposed method can tackle long horizon and sparse reward problems. As the paper discuss in introduction, learning in sparse reward environment is hard because it relies on the agent to enter the goal during exploration. However, the proposed approach seems only able to work in environments where exploration with random policy can generate trajectories that contain sufficient environment dynamics (e.g. dynamics near the goal states). How can the method learn that information without entering the goal?

Furthermore, it seems that the proposed approach only works for goal-oriented tasks (since we need to know the goal state for reward-shaping). I think this should be clearly stated in the paper.

There are some missing details which makes it difficult to draw conclusions:
1. How is the ‘success rate’ computed (e.g. in figure 7 and table 1).
2. How were the parameters selected (e.g. table 5 in the appendix). Why did you use the default the parameters?
3. How many runs are the curve averaged over and what’s the shaded region (e.g. one standard error)? Most of the results in the paper seem not statistically significant.
4. In figure 4, five domains are mentioned but only three of them are tested in the section 5.
5. Section 6.2 seems irrelevant to the paper. What’s the purpose of this section?
6. Figure 10 shows the result of using CPC feature directly vs. reward shaping. Are both feature using the same NN architecture, same PPO parameters, and same control setting? Moreover, the reward shaping method assumes we know the goal state but using CPC feature does not. Is it a fair comparison?

The paper has some imprecise parts:
1. The definition of MDPs (in section 2) is imprecise. For example, how is the expectation defined? how is the initial state sampled? What does $p\in\mathcal{P}$ (last line in the first paragraph) mean where $\mathcal{P}$ is the state transition function?

Minor comments which do not impact the score:
1. Figure 1 should come before figure 2.
2. It would have been better if there is a short description of how the hand-shaped rewards is designed for each domain in the main text.

[1] The Laplacian in RL: Learning Representations with Efficient Approximations


**Experience Assessment:**

I have published one or two papers in this area.

**Review Assessment: Checking Correctness Of Derivations And Theory:**

N/A

**Review Assessment: Checking Correctness Of Experiments:**

I assessed the sensibility of the experiments.

**Review Assessment: Thoroughness In Paper Reading:**

I read the paper thoroughly.

---

> ### Author Response · Authors · 2019-11-15
> **Response**
>
> Thank you very much for your insightful review. Below are our responses to your questions:
>
> 1. How is the ‘success rate’ computed (e.g. in figure 7 and table 1).
>
> “Success” in the grid word domains means the agent is able to reach the goal position (randomly set each run) within a certain time step limit (typically 100).
>
>
> 2. How were the parameters selected (e.g. table 5 in the appendix). Why did you use the default the parameters?
>
> The parameters were selected after hyperparameter searches. We did not exactly use any default parameters except for the architectural choice of using a GRU, which we found working quite well for all experiments.
>
>
> 3. How many runs are the curve averaged over and what’s the shaded region (e.g. one standard error)? Most of the results in the paper seem not statistically significant.
>
> Most of the experiments have 3 runs for each setting, and the shaded region represents one standard error. Our main goal was to compare our results against the standard setting (i.e. only with sparse rewards), and the comparisons for this pair are mostly statistically significant.
>
>
> 4. In figure 4, five domains are mentioned but only three of them are tested in the section 5.
>
> We decided to move 2 experiments to the appendix, and keep the 3 domains that best represent the power of the embedding.
>
>
> 5. Section 6.2 seems irrelevant to the paper. What’s the purpose of this section?
>
> This section aims to differentiate predictive coding from other unsupervised learning methods such as an autoencoder or VAE. Specifically, we wanted to show that predictive coding is more suitable for RL tasks than an autonecoder or VAE because the embedding can be trained to be background-agnostic. This is advantageous because in robotic tasks the background often will not stay stationary, and we could filter out less important information through predictive coding.
>
>
> 6. Figure 10 shows the result of using CPC feature directly vs. reward shaping. Are both feature using the same NN architecture, same PPO parameters, and same control setting? Moreover, the reward shaping method assumes we know the goal state but using CPC feature does not. Is it a fair comparison?
>
> Both settings use the same architectural and algorithmic settings. For both methods, the embedding is trained from states that include information about the goal (i.e. the goal is a part of the state), so features from both methods contain the goal states.

---

### Official Review · AnonReviewer2 · 2019-10-22
**Official Blind Review #2**

**Rating:** 3

**Review:**

The paper presents a method to derive shaping rewards from a representation learnt with CPC. They propose learning a CPC representation from some random data and fix it. They assume exploration is not too difficult so that rewards are achievable without additional mechanisms. Once a reward is achieved they can compute the embedding of the corresponding state as a goal under the CPC learnt representation either directly ie. distance or via a clustering step. The paper is clearly written and presents a nice idea. The paper is correct and part of the approach seems novel. I find the analysis of the results well executed though I think that they should be improved for publication.

Major points:
* I think this is a valid contribution and it seems like something the RL audience might be interested in.
* I am very surprised that CPC does such a good job given that the main object for learning in CPC is the distribution of trajectories which should be quite different in random exploration and the optimal policy. I presume this is because the environments are quite simple. This issue is of interest to the readers so an example of a simple failure case should make that point.
* Secondly, as nice as those embeddings in figure 5 look I wonder what happens in larger mazes with more structure i.e. somewhere where random walk will not be a uniform distribution and thus CPC will (most likely) not work as intended.
* The clustering bonus how do you prevent it from staying at the edge of the goal region and deriving infinite rewards from that ?
* Since these are not potential functions how do you prevent the rewards from biasing learning ? Ng et al. -- Policy invariance under reward transformations. This should be discussed in the paper because it is of practical importance.
* Contrastive learning has been used to find goal embeddings before Warde-Farley et al. Unsupervised control through non-parametric discriminative rewards. In that paper they don't need the CPC future state predictors but instead contrast the goal and the final state of the trajectory. They use the resulting embedding to learn a reward function and ignore the extrinsic reward. Interestingly, they show that the rewards can be learnt online (maybe ideas from that paper can be applied here).
Minor points:
* please make legends on plots readable size.


**Experience Assessment:**

I have published one or two papers in this area.

**Review Assessment: Checking Correctness Of Derivations And Theory:**

I assessed the sensibility of the derivations and theory.

**Review Assessment: Checking Correctness Of Experiments:**

I assessed the sensibility of the experiments.

**Review Assessment: Thoroughness In Paper Reading:**

I read the paper at least twice and used my best judgement in assessing the paper.

---

> ### Author Response · Authors · 2019-11-15
> **Response**
>
> Thank you very much for your insightful review. Below are our responses to some of your concerns:
>
> * The clustering bonus how do you prevent it from staying at the edge of the goal region and deriving infinite rewards from that ?
>
> The cluster reward is one time only. With the cluster reward, the MDP is no longer time-homogeneous, so we convert it to a time-homogeneous MDP by treating entering the cluster as the first goal, and entering the environment goal as the second goal.
>
>
> * Contrastive learning has been used to find goal embeddings before Warde-Farley et al. Unsupervised control through non-parametric discriminative rewards. In that paper they don't need the CPC future state predictors but instead contrast the goal and the final state of the trajectory. They use the resulting embedding to learn a reward function and ignore the extrinsic reward. Interestingly, they show that the rewards can be learnt online (maybe ideas from that paper can be applied here).
>
> Although our approach is currently off-line, we discussed near the end of our paper that converting the scheme to an online fashion is straightforward. Our goal in this paper is to focus on the properties of predictive embedding, and how they may contain information useful for RL; our contribution is largely on the finding that the predictive embedding flattens out the original state space and provide straightforward paths to any goal in the environment. We believe an online approach would work with our scheme, but that was not the focus of our paper.
>
>
> * I am very surprised that CPC does such a good job given that the main object for learning in CPC is the distribution of trajectories which should be quite different in random exploration and the optimal policy. I presume this is because the environments are quite simple. This issue is of interest to the readers so an example of a simple failure case should make that point.
> * Secondly, as nice as those embeddings in figure 5 look I wonder what happens in larger mazes with more structure i.e. somewhere where random walk will not be a uniform distribution and thus CPC will (most likely) not work as intended.
>
> With our scheme, having good exploration of the state space is key to training good predictive embedding. This is more challenging for more difficult environments such as a large maze. We believe that the need for good exploration is not unique to our approach; our focus of the paper is to provide a method to leverage good exploration data without reward signals, and this method is to train predictive embedding.
>
> For environments where random exploration is not enough (including the AntMaze environment), we overcame the issue by a) sampling from a robust distribution of initial states in the simulator b) training goal-conditioned policies for pure exploration strategies. These two approaches allowed us to collect good exploration data and train high-quality predictive embedding in more difficult environments
>
>
> * Since these are not potential functions how do you prevent the rewards from biasing learning ? Ng et al. -- Policy invariance under reward transformations. This should be discussed in the paper because it is of practical importance.
>
> A rigorous investigation of the mathematical properties of the latent space is out of the scope of this applicative paper. From an empirical perspective, however, the ability for an agent to learn an optimal policy in various experimental settings with our scheme reflect that the negative distance scheme is likely a potential function. Ultimately, this depends on whether the latent space induced by the encoder preserves the metric property of the original state space, and one could train an invertible map to enforce this.

---

### Official Review · AnonReviewer1 · 2019-10-23
**Official Blind Review #1**

**Rating:** 3

**Review:**

  *Synopsis*:
  This paper proposes using the features learned through Contrastive Predictive Coding as a means for reward shaping. Specifically, they propose to cluster the embedding using the clusters to provide feedback to the agent by applying a positive reward when the agent enters the goal cluster. In more complex domains they add another negative distance term of the embedding of the current state and goal state. Finally, they provide empirical evidence of their algorithm working in toy domains (such as four rooms and U-maze) as well as a set of control environments including AntMaze and Pendulum.

  Main Contributions:
  - Using the embedding learned through Contrastive Predictive Coding for reward shaping.
  - A reward shaping scheme that seems generally applicable to any embedding.


  *Review*:

  I think the paper provides a compelling motivation, and is well written. I think using the embedding learned through CPC could provide meaningful semantics for representation learning and for reward shaping (as done in the current paper), and encourage the authors to continue down this line of inquiry. Unfortunately, I have several concerns over the method as currently implemented and the empirical comparisons (specifically with the chosen competitors) which I detail below. Given these concerns I am unwilling to recommend accepting this paper, unless several of these concerns are addressed.

  1. This algorithm, by nature, is purely offline as the CPC and clustering all are currently done offline. Furthermore, the clustering portion of this approach requires states to be randomly sampled from the environment to create a nice set of clusters which are representative of the environment's full state space. These two requirements significantly limit this approach, especially when looking at domains where simulation is not possible, or the underlying state distribution is unknown. By and all, I don't think this means we should completely discount this method entirely and the authors do mention this as a detriment to their algorithm in section 6.3. I'm wondering if this paper should look at implementing an online version before publication, but think this is less prescient to the other concerns.

  2. I am concerned about the current policy learning scheme (i.e. tiered policy (1) go to the correct cluster (2) go to the goal) which seems to only be used by your approach. This invalidates the comparisons made with the other reward shaping schemes as this goes beyond reward shaping (you are learning two separate policies).

  3. The current competitors are unsatisfactory as they don't include other reward shaping techniques from the literature. Also the related works section seems to completely disregard this part of the literature. I would recommend comparing your approach to (methods you even cite!):
     - "Reward Shaping via Meta Learning" by Zou et. al https://arxiv.org/pdf/1901.09330.pdf
     - "Potential based reward shaping" by Gao et. al. https://www.aaai.org/ocs/index.php/IJCAI/IJCAI15/paper/viewPaper/10930
     (I'm sure there are others beyond these)

  4. I'm also concerned with the current significance of the results, particularly AntMaze, Half Cheetah, and Reacher. I'm especially concerned because I don't feel your competitors are not relevant/representative of the current state of reward shaping.

  5. It would be worthwhile to try this method on several RL algorithms (i.e. Q-learning, TRPO, SAC, DDPG, etc...). This will help readers understand if this approach is a general method, or only applicable to PPO.

  6. I quite like the idea of predictive coding (albeit the original scheme presented by Rao and Ballard 1999) as an unsupervised representation learning scheme, but am unsure this is critical for your method and the current approach is not really predictive coding in a sense (or at least the ideas I'm familiar with from cognitive computational neuroscience). I am concerned with the predictive coding idea being highlighted here as a key ingredient, but none of the papers containing the originating idea of predictive coding are mentioned. Rao and Ballard is one, but there is a rich literature following from this work into the free energy formulation (Friston) and active learning. These ideas should appear in your introduction, as you heavily rely on them. Also there are other predictive coding schemes for unsupervised representation learning (such as PredNets from David Cox https://coxlab.github.io/prednet/), which I believe should be mentioned. In light of these other methods, I'm not sure your discussion on only using predictive features for reward shaping is accurate, and instead these claims should be softened for only features learned through CPC.

  Missing experimental settings:
  - Number of runs tested
  - What are the error bars in your plots?

  I would also like to recommend "Deep Reinforcement Learning that Matters" by Henderson for a reference on how to conduct meaningful Deep RL experiments using policy gradient methods (https://www.aaai.org/ocs/index.php/AAAI/AAAI18/paper/viewPaper/16669). I think this paper could benefit from the recommendations made there.


  More questions/clarifications:

  Q1: What constitutes success in the grid world domains for table 1?

  Q2: If you were to train a single policy using your reward shaping scheme (i.e. not the tiered approach currently employed) how would the new policy perform? Are there problems with the current scheme where you have to have the tiered policy?

  Q3: Why CPC and not say an autoencoder or VAE? A comparison over many types of unsupervised embedding learning algorithms could be interesting and make your method more general than currently presented. It could also strengthen your argument for using CPC.

  Q4: How long were the trajectories for training CPC?

  *Other minor comments not taken into account in the review*
  - I think your labels are backwards in table 1.

**Experience Assessment:**

I have published one or two papers in this area.

**Review Assessment: Checking Correctness Of Derivations And Theory:**

I carefully checked the derivations and theory.

**Review Assessment: Checking Correctness Of Experiments:**

I carefully checked the experiments.

**Review Assessment: Thoroughness In Paper Reading:**

I read the paper thoroughly.

---

> ### Author Response · Authors · 2019-11-15
> **Response**
>
> Thank you very much for your insightful review. Below are our responses to your questions:
>
> Q1: What constitutes success in the grid world domains for table 1?
>
> “Success” in the grid word domains means the agent is able to reach the goal position (randomly set each run) within a certain step limit (typically 100). We have added this detail to the paper.
>
>
> Q2: If you were to train a single policy using your reward shaping scheme (i.e. not the tiered approach currently employed) how would the new policy perform? Are there problems with the current scheme where you have to have the tiered policy?
>
> A single policy works well for the negative distance scheme.
>
> For the clustering scheme, some simple changes to the state observations to account for whether the cluster reward has been received would allow the training of a single policy. An example is to just augment the state with a Boolean variable indicating whether the cluster reward has been received.
>
> Since the cluster reward is one time, the original MDP is no longer time-homogeneous, so a single policy naively learned with the added cluster reward will mistake the boundary of the cluster as a high reward zone.
>
>
> Q3: Why CPC and not say an autoencoder or VAE? A comparison over many types of unsupervised embedding learning algorithms could be interesting and make your method more general than currently presented. It could also strengthen your argument for using CPC.
>
> As we motivated in discussion, we found that predictive coding is more suitable for RL tasks than an autonecoder or VAE because the embedding can be trained to be background-agnostic, as predictive coding only extracts information from states that are helpful for summarizing the past and predicting the future. The goal of predictive information is to extract the valuable (for predicting the future) information from the past, rather than to reconstruct the samples. Autoencoders, in general, will reconstruct the entire scene (both background and robotic arm), which makes their latent space representations redundant and not task-specific.
>
>
> Q4: How long were the trajectories for training CPC?
>
> We have added this information to table 5 in the appendix. Each trajectory corresponds to one episode in the environment.

---

### Decision · Program_Chairs · 2019-12-19

**Decision:**

Reject

**Comment:**

The paper proposes to use the representation learned via CPC to do reward shaping via clustering the embedding and providing a reward based on the distance from the goal.

The reviewers point out some conceptual issues with the paper, the key one being that the method is contingent on a random policy being able to reach the goal, which is not true for difficult environments that the paper claims to be motivated by. One reviewer noted limited experiment runs and lack of comparisons with other reward shaping methods.

I recommend rejection, but hope the authors find the feedback helpful and submit a future version elsewhere.